# Interleukin-36 Cytokine/Receptor Signaling: A New Target for Tissue Fibrosis

**DOI:** 10.3390/ijms21186458

**Published:** 2020-09-04

**Authors:** Elaina Melton, Hongyu Qiu

**Affiliations:** Center for Molecular and Translational Medicine, Institute for Biomedical Sciences, Georgia State University, Atlanta, GA 30303, USA; emelton@gsu.edu

**Keywords:** IL-36R, IL-36, IL-38, IL-1 receptor family, tissue fibrosis, inflammation

## Abstract

Tissue fibrosis is a major unresolved medical problem, which impairs the function of various systems. The molecular mechanisms involved are poorly understood, which hinders the development of effective therapeutic strategies. Emerging evidence from recent studies indicates that interleukin 36 (IL-36) and the corresponding receptor (IL-36R), a newly-characterized cytokine/receptor signaling complex involved in immune-inflammation, play an important role in the pathogenesis of fibrosis in multiple tissues. This review focuses on recent experimental findings, which implicate IL-36R and its associated cytokines in different forms of organ fibrosis. Specifically, it outlines the molecular basis and biological function of IL-36R in normal cells and sums up the pathological role in the development of fibrosis in the lung, kidney, heart, intestine, and pancreas. We also summarize the new progress in the IL-36/IL-36R-related mechanisms involved in tissue fibrosis and enclose the potential of IL-36R inhibition as a therapeutic strategy to combat pro-fibrotic pathologies. Given its high association with disease, gaining new insight into the immuno-mechanisms that contribute to tissue fibrosis could have a significant impact on human health.

## 1. Introduction

Tissue fibrosis is a common pathological alteration observed in multiple organs and a major unresolved medical problem resulting in increased morbidity and mortality [1,2]. Acute or chronic stimuli, such as autoimmune reactions, infections, mechanical injury, and age-related degradation, result in an excessive deposition of extracellular matrix (ECM), which replaces the damaged tissues, leading to the disturbance of the original healthy tissue architecture and function [2,3]. The molecular mechanisms involved in the advancement of tissue fibrosis are not entirely understood, which delays the advancement of effective anti-fibrotic therapeutics. Moreover, understanding the molecular triggers that cause fibrosis of tissues under various pathological settings is an important question to address in order to find new ways to treat this condition.

Because of its vast role in disease progression and involvement in inducing pro-fibrotic processes in multiple cell types, inflammation has developed into a potential target for anti-fibrotic therapies. Given the observations that transforming growth factor beta (TGF-β) induces fibroblast activation, fibrosis is associated with TGF-β polymorphisms, and TGF-β expression is highly correlated with several types of organ fibrosis, its inhibition has been tested as a therapeutic strategy in numerous studies [2,4,5,6,7,8,9]. The use of pharmacological inhibitors and antibodies of TGF-β resulted in promising outcomes; however, these approaches are not widely accepted as therapies to treat fibrosis [10,11]. To this end, many inflammatory molecules in addition to TGF-β have been identified as targets for eliminating fibrosis. For example, interleukin 10 (IL-10), an anti-inflammatory cytokine, was found to suppress the formation of lung fibrosis in vivo, yielding promising leads to eradicate fibrogenesis [12,13].

The interleukin-1 family (IL-1 family), a group of 11 cytokines that play a central role in the regulation of immune and inflammatory, has been found to be involved in fibrotic pathogenesis. IL-1α and IL-1β are two members of IL-1 family that were first discovered to possess strong pro-inflammatory effects. Nine other cytokines, IL-1Ra, IL-18, IL-33, IL-36Ra, IL-36α, IL-36β, IL-36γ, IL-37, and IL-38, have been included in the IL-1 superfamily owing to their structural similarities and the overlap in their function and their binding receptors (IL-1 receptor family). The nomenclature of this receptor family was recently established to reflect the order in which they were discovered. The IL-1R family consists of 10 members (IL-1R1–IL-1R10). Each member of the IL-1 receptor family has distinct expression patterns, agonists, and biological functions, likely contributing to their differential roles in regulating cellular inflammatory responses [14]. Given their essential cellular functions, the IL-1 family is considered an integral component of inflammation and many of its members have been linked to an array of inflammation-driven diseases, including fibrosis [15,16,17]. For example, IL-33 and its receptor ST2 (also known as IL-1R4) have been implicated in the fibrotic pathology of several models of fibrosis [18,19]. In addition, recent studies provide mounting in vivo and in vitro evidence indicating that IL-36 and its corresponding receptor IL-36R (also known as IL-1R6) mediated signaling promote the biological processes governing fibrosis, which, in some instances, leads to organ impairment or failure [20,21]. As newly characterized members of the IL-1 family of receptors and cytokines, IL-36/IL-36R has forged new discoveries in the immune pathways associated with psoriasis, arthritis, obesity, intestinal, and many other chronic diseases [22,23,24,25], making it an attractive target for immune-based processes involved in disorders such as fibrosis.

In this review, we highlight the most recent works that examine the fibrogenic role of IL-36/IL-36R in several models of disease such as lung, kidney, cardiac, intestinal, and pancreatic fibrosis. We also summarize the new insights in the IL-36/IL-36R- related mechanisms involved in tissue fibrosis and the potential of IL-36R inhibition as a therapeutic strategy to combat pro-fibrotic pathologies. Although more details need to be uncovered, this review provides novel information that would escalate the understanding of the pathogenesis of multiple diseases and stimulate more investigation in this field.

## 2. Molecular Basis and Biological Function of IL-36R in Cells

The newly discovered IL-36R has structural and functional similarities to the IL-1R family of receptors. These features allow extracellular signals to bring about immune responses in various cell types. In this capacity, IL-36R signaling initiates a biochemical pathway that leads to inflammation.

### 2.1. IL-36R Molecular Structure

The human and rat *IL-36R* gene, also known as IL-1R6, interleukin receptor related protein 2 (IL-1Rrp2) or interleukin 1 receptor like 2 (IL-1RL2), was the sixth IL-1 receptor identified after the first (IL-1R1) was discovered in 1988 [16]. It was first cloned and characterized by Lovenberg et al. in 1996 using a homology-based cloning method [26]. This 561 AA protein has a molecular weight estimated to be 65 kilo dalton (KD) and shares 42% identity with the IL-1 receptor. The gene is located on chromosome 2 and its sequence is conserved in monkey, dog, cow, mouse, rat, and zebrafish [27,28]. Biochemical analysis has revealed that the IL-1R family members share a similar molecular structure that consists of three structural domains with distinct functions, which help these receptors facilitate their biological activity [29]. This finding was further corroborated by recent X-ray crystallography data reporting the crystal structure of IL-36R bound to a monoclonal antibody, B1655130 Fab [30]. As shown in Figure 1, the common domains shared by this group of receptors include an extracellular domain, a transmembrane region, and a Toll-IL-1 receptor domain (TIR). The extracellular region contains three Immunoglobulin (Ig) domains (D1, D2, and D3). This stretch of amino acid sequence is important for cytokine recognition. The TIR domain, which resides in the cytoplasm, is essential for co-receptor binding and signal transduction (Figure 1). This domain has a sequence identity similar to that found in toll-like receptors (TLRs), which are also important mediators of inflammation [15].

### 2.2. IL-36R Function-Associated Molecules

Studies indicate that IL-36R confers its function through interacting with other molecules to constitute an assembly of the receptor/ligand/adaptor complex, which is regulated by both specific agonists and antagonists.

Biochemical studies have revealed that IL-36R functions as a heterodimer that associates with a co-receptor called interleukin-1 receptor 3 (IL-1R3), also known as interleukin 1 receptor accessory protein (IL-1RAcP) [31]. It has also been shown that IL-36R stimulates a signal transduction event through a mechanism that involves the actions of an adaptor protein myeloid differentiation primary response gene 88 (MyD88) and the interleukin-1 receptor associated kinase (IRAK) (Figure 1) [32,33,34]. In addition, several known agonists (IL-36α, IL-36β, and IL-36γ) and antagonists (IL-36Ra, IL-38) have been shown to modulate its activity [22,35,36]. Signal transduction through this pathway leads to mitogen-activated protein kinase (MAPK) induced Activated Protein-1 (AP) and nuclear factor kappa-light-chain-enhancer of activated B cells (NF_K_B) dependent upregulation of pro-inflammatory gene expression (Figure 1) [31,37].

In 2016, Kao and colleagues showed that the IL-36R/IL-36/IL-1RAcP complex could be disrupted by mutations in the extracellular domain, demonstrating its importance in the assembly and signaling function of the receptor [38]. Additionally, this group showed that glycosylation of the asparagine residues in IL-36R is an important factor in IL-36/IL-36R signaling, as well as its trafficking to the plasma membrane [38]. Posttranslational modification of the IL-36R cytokines (IL-36α, IL-36β, and IL-36γ) is also crucial for the biological activity of the IL-36/IL-36R complex. Similar to IL-1β and IL-18 cytokines, the IL-36 cytokines must first be processed by caspases in order to become active. The precursor form of IL-36, pro-IL-36α, is cleaved by either catalase G or elastase. Catalase G is also involved in processing pro-IL-36β, while three distinct enzymes, Elastase, proteinase, and catalase S, act on pro-IL-36γ to convert it into its active mature form [36].

### 2.3. Distribution and Biological Functions of IL-36R

IL-36R is highly expressed in the skin and epithelial cells, which are the major cell types involved in skin psoriasis [22]. According to the RNA-seq alignment reported by BioProject: (PRJEB4337), the human isoform of *IL-36R* is highly expressed in the esophagus, thyroid, kidney, skin, adrenal gland, and the gall bladder, yet the roles of IL-36R in these tissues are not fully defined [28].

IL-36R signaling has been shown to influence the cellular inflammatory response in multiple ways, which has been nicely reviewed by Zhi-Chao Yuan et al. 2019 [22]. The multifaceted mechanism attributed to IL-36R includes inducing cytokine production in cells such as skin keratinocytes, endothelial cells, and lung fibroblasts. It also produces pro-inflammatory responses in immune cells. For example, microarray analysis showed that IL-36β treatment stimulated an induction of cytokines and chemokines (namely, granulocyte colony-stimulating factor (*G-CSF*), interleukin 17C (*IL-17C*), C-C motif chemokine ligand 20 (*CCL20*), and interleukin 8 (*IL-8*)) in normal primary keratinocytes isolated from humans [35]. Similar pro-inflammatory effects were observed in endothelial cells, which exhibited elevated CCL20, C-C motif chemokine ligand 2 (CCL2), IL-8, and vascular cell adhesion molecule 1 (VCAM 1) levels when stimulated with IL-36γ [39]. Likewise, in healthy lung fibroblasts, upregulation of pro-inflammatory markers such IL-8 and CCL20 was potentiated through the activation of the IL-36R pathway via IL-36γ [40]. IL-36R associated pro-inflammatory effects has also been demonstrated in cell types such as dendritic cells, macrophages, human leukemic monocytes (THP-1 cells), neurons, and glial cells [41]. In addition to aiding in the production of cytokines, IL-36R signaling also contributes to inflammation by modulating lymphocyte activity. This membrane receptor is expressed on a variety of leukocytes that have the capacity to respond to IL-36 cytokine stimulation. For example, IL-36 treatment can provoke an immune response in dendritic and T cells, as well as stimulate their differentiation. [42,43,44,45].

Given its tissue expression profile and cellular activities, it is plausible to propose that IL-36R expression may be directly related to its function in regulating inflammation in corresponding tissues. For example, loss of function and inhibitor studies (employing antagonists IL-36Ra or IL-38) have shown that blocking IL-36R signaling can alleviate several inflammatory diseases, thus showing promise that IL-36R and its ligands are potential targets for preventing inflammatory pathologies [22]. IL-36R has been studied in several models of inflammation, such as arthritis, lupus, and psoriasis. Included in these studies were reports that, in the absence of IL-36R, pathologies related to psoriasis inflammation are diminished [46,47,48]. Another report corroborated this finding by providing evidence that dampening the repressive role of IL-36Ra on IL-36R can heighten the pathological processes involved in psoriasis progression [47]. On the contrary, overexpression of an IL-36R stimulatory ligand such as IL-36α can lead to the formation of skin inflammation [49].

In summary, as shown in Figure 1, the assembly of IL-36R/IL-36/IL-1R3 complex can initiate a culmination of signaling events that lead to the generation of pro-inflammatory responses by a variety of cell types. Moreover, the distinct IL-36 induced response is likely dependent on the cell type and tissue environment.

## 3. Cellular and Molecular Mechanisms of Tissue Fibrosis

### 3.1. Overview of Tissue Fibrosis Development

Fibrosis is a process in which thick fibrillar scars form as a result of tissue injury or trauma. Fibrosis is generally characterized by (1) an accumulation of immune cells; (2) the presence of extracellular matrix (ECM) and fibrotic features in parenchymal tissue; and (3) the phenotypic change of resident cells to “myofibroblasts like” cells [2]. Following a tissue injury or an event that leads to cellular damage, the wound healing process begins with the recruitment of immune cells. Recent reviews of the literature on this topic have found that this process involves several immune cell types, including the following: macrophages, T cells, eosinophils, basophils, and lymphoid cells of group 2 (ILC-2), all of which contribute to the inflammatory events of fibrosis [50,51,52,53]. The macrophages produce inflammatory signals, such as transforming growth factor beta (TGF-β), which stimulate the expression of alpha smooth muscle actin (α-SMA) in fibroblasts via an Mothers against decapentaplegic (SMAD) dependent pathway [3,54,55,56,57], and release of the collagen [1,2]. It has been shown that, during fibrosis, resident epithelial, endothelial, and smooth muscle cells change their phenotype to resemble myofibroblasts [2,58,59,60,61,62]. The phenotypic switch in epithelial and endothelial cells to myofibroblasts like cells is called epithelial to mesenchymal transformation (EMT) and endothelial to mesenchymal transformation (EndMT), respectively. External sources of myofibroblasts precursors also play a role in this process. For instance, circulating hematopoietic cells can differentiate into fibrocytes upon stimulation by TGF-β, endothelin-1 (ET), IL-13, and GM-CSF, while pericytes have been found to transition into myofibroblast “like” cells after TGF-β, platelet-derived growth factor (PDGF), and IL-1β stimulation [51,63,64,65,66,67,68]. Collectively, the converted resident cells and myofibroblasts derived from external sources aid in the development of fibrosis by producing and releasing ECM factors at the injury site (Figure 2). It is also believed that the ECM itself impacts the development of fibrosis [69]. This notion was supported by data showing that fibrotic extracellular matrix induces the expression of ECM proteins by reducing the expression of miR-29, which negatively regulates the transcription of stromal genes responsible for ECM enrichment. Reduction of miR-29 is thought to contribute to the pathological progression of fibrosis by driving the upregulation of ECM factors and promoting ECM synthesis [70].

Various subsets of T cells have also been linked to fibrosis development [50]. Several studies, for example, found that T helper 2 cells (Th2 T) cells produce IL-4, IL-13, and IL-5, which act on macrophages to induce the production of pro-fibrotic cytokines and growth factors (TGF-β and PDGF) that lead to fibrosis [1,71,72]. Interestingly, resident epithelial and endothelial cells can induce the IL-4/IL-13/IL-5 mediated response in Th2 cells via release of alarmins such as IL-25, IL-33, and thymic stromal lymphopoietin (TSLP) [51,73]. PDGF released from Treg T cells is also associated with elevated activation of fibroblasts and fibrosis formation in the lung [74]. Cortez et al. and others found that IL-17 produced by Th17 cells can induce the upregulation of inflammatory and ECM related genes (IL-6, GM-CSF, TNFα, C-X-C motif chemokine ligand 1 (CXCL1), CXCL2, CCL20, and MMPs) in resident tissue cells, likely through an NF_K_B/ CCAAT/Enhancer-Binding Protein Beta (CEBPβ)/AP-1-dependent mechanism [75,76,77]. Contrary to the pro-fibrotic activities of T cells, Th2 production of IL-10 has been shown to have anti-fibrotic effects and been linked to tissue repair [78]. Similarly, IFNγ released by the Th1 subclass of T cells also has inhibitory effects that can help mitigate TGF-induced ECM production and fibrosis [79,80]. Taken together, inflammatory cytokines and growth factors (namely, interleukins, TGF-β, PDGF) produced by the recruited immune cells play a critical role in the progression of fibrosis. In addition, myofibroblasts derived from resident or nonresident cells likely contribute to the accumulation of collagen and ECM factors that lead to fibrosis. The major cells and cytokines involved in this process are summarized in Figure 2.

The wound healing process also includes a fibrosis resolution step wherein scar tissue is remodeled, parenchymal cells are regenerated, and the actions of myofibroblasts are halted. The fibrotic tissue remodeling phase of fibrosis reversal is regulated by metalloproteinases (MMPs), which functions to decompose ECM through their degradative action on collagen [54]. A number of studies have shown that overexpression of MMPs reduces tissue fibrosis. For example, overexpression of MMP1 resulted in reduced fibrosis in models of liver and cardiac fibrosis [81,82]. Restoring parenchymal cells in tissues augmented by fibrotic tissue is an important step in resolving fibrosis and, more importantly, it is essential to regaining normal organ function. Several cellular mechanisms are operational to control the presence of ECM producing myofibroblasts in fibrotic tissues. They include cell death through apoptosis, cell cycle arrest by senescence, or conversion of myofibroblasts back to their original phenotypic state. The benefits in inducing these pathways to reverse fibrosis are nicely reviewed by Jun and Lau in a 2018 review article [54,83].

Under conditions in which fibrotic scars cannot be resolved or remodeled, fibrosis can persist especially in a favorable environment that allows fibrosis to continue to manifest. Uncontrolled induction of fibrosis can lead to fibrotic scar tissue formation and even mechanical changes that impact the stiffness and function of the tissue. The effects of tissue stiffness on fibrosis progression can be investigated by mechanically stimulating cells to emulate stiffness. In one such study, mechanical stimulation of fibroblasts led to the activation of the Hippo pathway, which induced an enrichment of ECM contents driven by transcriptional modulators of pro-fibrotic events: YAP (Yes-associated protein) and TAZ (transcriptional co-activator with a PDZ binding domain) [84]. Furthermore, there is substantial data supporting the notion that non-functional fibrosis resolution creates a situation that promotes fibrosis and further complicates disease [69,85]. Thus, understanding the pathways involved is critical for preventing the malfunction of organs owing to the unresolved damaging effects of tissue thickening, stiffness, and augmentation of cellular morphology.

### 3.2. The Pathological Role of IL-36/IL-36R in the Development of Tissue Fibrosis

Much progress has been established regarding defining the role of IL-36R in additional immune-related diseases. It has been demonstrated that a disease called DITRA (deficiency of the IL-36R antagonist) can develop in patients expressing an inactive form of IL-36Ra. This gene deficiency causes serve psoriasis in patients affected by DITRA [47]. In a recent publication, IL-36R has been linked to the metabolic syndrome. This work found that inhibition of IL-36R can prevent diet-induced obesity and metabolic orders [86].

The idea that suppressing the IL-36R pathway can prevent fibrotic diseases has also gained traction. Scheibe et al. 2019 provided evidence that blocking IL-36R decreased the progression of fibrosis in a model of chronic intestinal inflammation [87]. Subsequent studies in various models of fibrosis later revealed additional evidence of a possible molecular link between IL-36R and fibrogenesis. In recent years, IL-36R has gained much attention in the field of fibrosis, owing to the discovery of IL-36R-functional associated molecules such as its agonists (IL-36α, β, and γ) and antagonists (IL-36Ra and IL-38), which are involved in the fibrosis of multiple tissues such as lung, kidneys, and heart [15,16,21,22,88,89,90]. Taken together, this body of work points out a new avenue of research centered on understanding the mechanisms involved in the IL-36R mediated fibrosis progression, which we will review here.

#### 3.2.1. Stimulation of IL-36R Induces Lung Fibrosis

Lung fibrosis is characterized by inflammation, collagen buildup, and scaring of the pulmonary tissue [88]. As a result of scar tissue accumulation, respiratory complications and lung disease can develop. Much evidence suggests that inflammatory pathways that include members of the IL-1 cytokine family mediate this process [88,91,92]. Recent studies have shown that IL-36R ligands are implicated in the immunopathology of lung inflammation caused by asthma or cigarette smoking [40,93,94,95]. In line with these studies, IL-36 administration was associated with an influx of neutrophils to the lung tissue and an increase in Cdllc^+^ macrophage activation of CD4 T cells, thus confirming its involvement with lung immune-physiological processes [43].

The role of IL-36 in lung fibrosis was confirmed in a recent study, which showed novel determinants of tissue repair and fibrosis [90]. Using a bioinformatics approach, they found that fibrotic tissue from mice subjected to the implantation of a foreign biomaterial shared a similar subset of unique macrophages (CD9^+^ IL-36γ^+^) with those isolated from fibrotic lung tissue. This subgroup of macrophages was also enriched with markers for inflammation including IL-17A and IL-17Ra (receptor for IL-17A). The observation that IL-17A is upregulated is consistent with previous reports showing that IL-17A is a contributing factor to the molecular pathways that regulate pulmonary fibrosis [96,97]. Moreover, this work not only validated the important role of IL-17A, but also supports the claim that IL-36R is a contributing factor in lung fibrosis. The molecular mechanisms linking in IL-36γ/IL-36R and IL-17 to lung fibrosis have yet to be discovered. On the basis of these studies, one can speculate that, in the context of lung fibrosis, IL-36 producing macrophages activate Th17 T cells, which consequentially leads to fibrosis.

#### 3.2.2. Enhanced IL-36R Ligand is Associated with Kidney Fibrosis

Renal fibrosis often coincides with inflammation, a buildup of collagen factors, and an alteration in the function and physiology of the nephron [98]. The accrual of collagen and fibronectin, both structural components of the ECM, has been named in connection with fibrosis development. The consequences of renal fibrosis include disruption of proper kidney function and eventually renal failure. The pro-fibrogenic events that advance the progression of fibrosis have been linked to renal inflammation, fibroblast activation, glomerulotubular disconnection, and genetic factors [98]. Renal inflammation has been subjected to much evaluation, and has gained attention in the field of nephrology because of its high association with renal disease. According to these studies, IL-36R ligands appear to be highly associated with nephron disease progression and fibrosis, making this pathway a likely candidate for causing advancement in disease. Early work by Ichii described the observation that the development of tubulointerstitial lesions (TILs) was coupled with a robust elevation of IL-36α expression in the mouse model of glomerulonephritis [99]. Known characteristics of TILs include tubular dilations, immune cell infiltration, phenotypic switch of tubular epithelial cells, and interstitial fibrosis.

Recent evidence shows that the IL-36R signaling pathway mediates several contributing factors of kidney fibrosis, thus highlighting the important role of IL-36R in fibrogenic renal pathology. Chi et al. 2017 nicely describes the involvement of IL-36 in renal tubulointerstitial lesion (TIL) formation, which is a hallmark of renal fibrosis [21]. Renal tissues from patients with TILs exhibited higher IL-36α expression compared with healthy subjects. Enhanced renal fibrosis scores measured and detected by immunohistochemistry (IHC) paralleled the upregulation of IL-36α. An elevated level of renal IL-36α was also observed in the renal tubular epithelial cells (TECs) of the unilateral ureteral obstruction (UUO) mouse model of renal fibrosis. This enhanced IL-36α expression is also apparent in additional pathological conditions that are associated with fibrosis, such as chronic kidney disease (CKD) and acute kidney injury (AKI) [21,100]. Thus, collectively, these studies fortify the notion that the IL-36R pathway plays a significant role in kidney pathology.

Notably, IL-36 knockout (KO) studies described in Ichii et al. 2017 provide additional insight into how IL-36 functions in the setting of renal fibrosis. This group observed evidence of increased interstitial inflammation, cell death markers, α-SMA^+^myofibroblasts presence, TILs, and areas of fibrosis in the distal tubules (DT) of kidneys isolated from the UUO mouse [101]. Most importantly, they found that IL-36α was dramatically overexpressed in the kidney of the UUO mouse model. This finding is consistent with the work of Chi et al. 2017, who reported similar results. They also found that IL-36α expression was highly correlated with increased blood urea nitrogen (BUN) and creatine levels, which are measures of renal function. In line with this discovery, they were able to show a connection between IL-36α expression; biomarkers of TIL, namely kidney injury molecule -1 (Kim-1); and renal dysfunction in the acute kidney injury (AKI) model [101]. The fibrosis-induced upregulation of Kim-1 was substantiated by a subsequent RNA seq analysis, which showed that *Kim-1* expression was elevated in the phosphate induced kidney injury model [102]. Thus far, the relationship between IL-36α expression and Kim-1 is unknown. Most interestingly, IL-36R KO studies in the UUO mice revealed that, when compared with wild type mice, IL-36R KO mice exhibited improved renal function, decreased fibrosis, and reduced tissue inflammation, supporting the notion that inhibiting IL-36R activity can rescue mice from the fibrosis phenotype. Moreover, this work further implies that IL-36R may be involved in molecular pathways that influence fibrosis in the setting of kidney injury.

#### 3.2.3. Repression of IL-36R Protects against Cardiac Fibrosis

The functional and morphological consequences of myocardial infarction have been highly targeted areas in the field of heart disease. Fibrosis developed after myocardial infarction (MI) impairs the contractile capability, leading to heart failure (HF) [89]. Studies have shown that immune responses drive many of the processes that influence cardiac function in MI [103]. Aberrant side effects such as cardiac inflammation can lead to “ventricular remodeling” following an MI event [103,104]. For example, inflammatory factors such IL-1 cytokines have recently been considered as pro-fibrotic mediators of fibrogenesis after MI. In addition, sIL-33 (soluble form of interleukin −33) has been shown to generate fibrotic phenotypes in myocardial tissue [105,106]. Reciprocally, when inflammation is inhibited, the cardiovascular physiological changes such as ventricular remodeling are minimized [104,107,108].

The presence of IL-38 (an IL-36R inhibitor) in MI patients has also drawn attention to the role of the IL- 1 family as functional determinants of cardiac function [109]. Wei et al. followed up their 2015 investigation of IL-38′s contribution to MI pathology with subsequent in vivo mouse studies, which showed that treatment with recombinant IL-38 resulted in a disruption of cardiac fibrosis and improved ventricular function [89]. This observation reinforced the significance of IL-36R and its antagonists in the development of cardiac fibrosis.

#### 3.2.4. Inhibition of IL-36R Activity Prevents Intestinal Fibrosis

According to endoscopy examinations and histological reports, there is an association between inflammatory bowel diseases (IBDs) and the occurrence of fibrosis. There are several studies that found that, upon the onset of Crohns disease (CD) and ulcerative colitis (UC), fibrotic changes appear in intestinal and bowel tissues [110,111,112,113]. It is known that intestinal fibrotic matter such as collagen, laminan, and fibronectin are produced by activated myofibroblasts that contribute to fibrotic scarring [114]. In line with these studies, a study showed that fibrogenesis genes such as *Col1a1* (collagen type I alpha), *Col1a2* (collagen type I alpha 3), *Col3a1* (collagen type 3 alpha 1), and lumican (a collagen-associated proteoglycan) are induced in chronic colitis. The enhanced expression of fibrogenesis genes correlated with the disease stage and was highly associated with robust levels of proinflammatory mediators [115]. The buildup of fibrosis in the intestine can cause strictures, which are regions of narrowed intestine or colon. Strictures are a pathophysiology consequence of CD and UC development in patients [116].

Elevated levels of IL-36 cytokines in models of CD and UC provided substantial evidence that IL-36R signaling may be inter-related to the immuno-pathology of intestinal diseases [117,118]. Recently, Scheibe et al., 2019, found that macrophages expressing IL-36α, CD14, CD64, and CD163 were localized to areas of fibrosis in mice subjected to CD, thus implicating IL-36α as a possible modulator of fibrosis. Subsequent studies in mouse models of IBD showed that depleting IL-36R genetically or with antibodies that inhibit IL-36R activity prevented the intestinal fibrosis progression observed in the wild type or untreated controls. Specifically, in the case of CD, IL-36R KO mice had a lower fibrosis score and displayed reduced submucosal thickening [87]. This work provided compelling evidence that confirmed that blocking IL-36R with an IL-36R antibody can result in less fibrosis and reduce the abundance of activated fibroblasts (α-SMA^+^) in the UC mouse model. Most notably, this work displayed the therapeutic capacity of IL-36R inhibition as a method to treat fibrostenosis in a mouse model of intestinal colitis. The feasibility of using IL-36R inhibition as method to disrupt fibrosis in human IBD cases is unknown. Currently, there are ongoing clinical trials accessing the inhibitory effects of an IL-36R antibody called BI655130 on UC [119]. Although the clinical trial has been initiated (https://clinicaltrials.gov/ct2/show/NCT03482635), the results are not yet reported. It will be interesting to see if blocking IL-36R will also have an effect on fibrosis in the context of IBD conditions in human subjects.

#### 3.2.5. IL-36/IL-36R Signaling is Involved in Pancreatic Fibrosis

The association between pancreatitis and the incidence of IL-36/IL-36R mediated fibrosis is a new area of research. Most recently, it was found that expression of the IL-36R pathway components may have some significance in the development of pancreatic fibrosis. In this study, both IL-36R and IL-36α were expressed in the pancreatic tissue obstructed with fibrotic ECM deposition. It was discovered that IL-36α treatment can stimulate the expression of CXCL1, CXCL8, MMP-1, and MMP-3 in myofibroblasts isolated from the human pancreas. Mechanistic studies indicated that the IL-36α induced upregulation of fibrogenic gene expression was MAPK- and NF_K_B-dependent [120]. In vivo IL-36R knockout studies would further strengthen the hypothesis that IL-36R plays a role in pancreatic fibrosis progression. From the work conducted thus far, one could derive a possible mechanistic model in which IL-36α mediates the upregulation of inflammatory and fibrosis molecules through a mechanism that depends on MAPK and NF_K_B activity. The relative studies linking IL-36/IL-36R to fibrosis are summarized in Table 1.

### 3.3. Molecular Mechanisms of IL-36R Mediated Fibrosis

#### 3.3.1. IL-36R Promotes Fibrosis via Regulation of Immune Cell Responses

Despite the differences of fibrosis among the tissues mentioned above, studies indicated that IL-36/IL-36R may influence fibrosis progression through multiple common mechanisms (Figure 3). One possible mechanism may include dendritic cells (DCs), which function to regulate the activation and differentiation of T cells, driving inflammation and fibrosis.

Evidence of this mechanism is apparent in the cardiac fibrosis model. Wei and colleagues found that IL-38, an IL-36R inhibitor, reduced immune cell (neutrophil and macrophage) presence after MI [89]. In vitro studies revealed that IL-38 prevents DCs from maturing into CD40^+^ CD86^+^ MHC-II (major histocompatibility complex class II)^+^ cells and impedes their ability to produce pro-inflammatory cytokines IL-23, TNFα (tumor necrosis factor alpha) and IFNγ (interferon gamma) [89]. It is well established that these cytokines are linked to leukocyte recruitment and activation [121,122,123,124]. It was also reported that IL-38 treatment downregulated CD4 T cells accumulation, but upregulated the regulatory T cell subgroup called Foxp3+ Treg, which can repress inflammation and inhibit remodeling [89,125]. In summary, the current data implies that IL-36 mediates cardiac fibrogenesis in a comprehensive mechanism by inducing the maturation of DCs, increasing the activation of CD4^+^T cells, and reducing the population of T regulatory cells.

In alignment with the discoveries in cardiac tissue, the involvement of the IL-36 mediated dendritic and T cell associated mechanism has also been substantiated in kidney fibrosis. It was reported that silencing IL-36R in the UUO mouse of renal fibrosis led to the reduction of inflammation, which was displayed by less tissue accumulation of leukocytes (i.e., macrophages and T cells). It was suggested that this could be explained by the reduction of Nod like receptor family pyrin domain containing 3 (NLRP3), IL-1β, and caspase 1 levels observed in UUO mice lacking IL-36R. The NLRP3 mediated production and release of IL-1β and IL-18 is a well-recognized proponent of the pro-inflammatory response pathway in many cell types [126]. In addition, they showed that the NLRP3 inflammasome activation was diminished in IL-36α treated cells lacking MyD88, thus providing evidence that IL-36 mediated inflammasome activation is MyD88 dependent [21]. This finding was corroborated by a subsequent study demonstrating a possible link between IL-36α expression, renal fibrosis, tubular damage, and NLRP3 inflammasome activation in a dietary phosphate induced renal injury mouse model [127]. The impact of IL-36R suppression on T cell function was also evaluated in the kidney model of fibrosis [21]. Several lines of evidence have led to the view that T cell activity is involved in renal inflammation and fibrosis [128,129]. Most notably, the suppression of IL-36R signaling eliminated fibrosis formation and diminished T cell responses (i.e., IL-23 and IL-17A production) in models of kidney disease. In contrast, IL-36 stimulation initiated the elevation of RORγt (a transcription factor that regulates Th17 T cell function). Interestingly, the upregulation of RORγt levels was matched by a rise in Th17 T cell differentiation [21]. Hence, these findings show a possible link between IL-36R/IL-17/T cell differentiation and fibrosis.

Additionally, studies in lung fibrosis have further verified the contribution of T cell function in IL-36R mediated fibrosis. Treating mice with a lentivirus expressing recombinant IL-38 reduces the accumulation of lung leukocytes [88] and tissue cytokine production. Of particular interest was the observation that IL-1β and IL-17A were affected by the overexpression of IL-38. In line with this, the blockade of IL-17A production by IL-38 implies that Th17 T cell activation may be an important governing element of IL-36 induced lung fibrosis. This speculation was affirmed by another study reporting that IL-38 can block IL-36 induced T cell IL-17A and IL-22 production [130]. IL-17A and IL-22 have been shown to be pivotal mediators in inflammatory diseases, including disease states that lead to fibrosis [88,92,131]. These data indicate that, perhaps, in the diseased lung, IL-38 is induced to protect the lung tissue from fibrosis caused by inflammation [88]. According to a 2018 study, this postulate may hold true in human fibrosis, as IL-38 was evaluated in human lung samples from patients exhibiting acute idiopathic pulmonary fibrosis [132].

Given that T cell activation was observed in cardiac, kidney, and lung models of fibrosis, this pathway may be a common thread among specific types of organ fibrosis. One could surmise the idea that the mechanism by which IL-36R mediates inflammation and fibrosis likely involves NLRP3 and the IL-23/IL-17 immune response pathways (Figure 3). The involvement of IL-36 induced Th17 T cell activity in other examples of organ fibrosis has yet to be determined. These data also point out that the activity of immune cells (i.e., DCs and T cells) is likely inter-connected in the setting of fibrosis and the nature of this interaction may be dependent on the tissue type and the disease state.

#### 3.3.2. IL-36R Promotes Fibrosis by Modulating Fibrogenic Factors in Fibroblasts

A second mechanism that might be operational in the setting of tissue fibrosis is one in which IL-36R signaling induces the activation of fibroblasts to myofibroblasts, and drives the expression of fibrogenic factors (Figure 3). Current experimental data suggest that IL-36 cytokines may promote tissue fibrosis by stimulating fibroblasts to produce collagen and pro-inflammatory mediators through a MyD88 dependent mechanism. This was demonstrated by both intestinal and pancreatic fibrosis models, which showed that stimulating fibroblasts with IL-36α induced inflammation and fibrosis genes (namely, CXCL1, MMP1, MMP3, MMP10, MMP13) and increased collagen type VI expression. Interestingly, the elevated gene expression of the fibrosis pathway was dependent on MyD88 activity [87]. This work corroborates an earlier study showing that the uptick of fibrogenic genes was downregulated by NF_K_B inhibition, which is a downstream effector of MyD88 signaling [115,120]. Taken together, these studies suggest that, in fibroblasts, IL-36R controls fibrosis via the upregulation of the fibrogenic program through a MyD88 dependent pathway.

#### 3.3.3. IL-36R Induces Tissue Fibrosis through Regulating Enzyme-Mediated Collagen Remodeling

Other possible IL-36R related mechanisms that may aid in the development of fibrosis are also attracting attention. In 2017, Ichii reported another likely mechanism of IL-36R driven renal fibrosis. Most noteworthy was the discovery that renal tissue from IL-36 KO mice showed a marked increase in a collagen degradation enzyme called inactive serine protease 35 (Prss35) and altered the expression of many sensory proteins, including those in the olfactory (Olfr) and vomeronasal (Vmnr) receptor families [101]. Prss35 activity has been shown to be important for fibrosis, while Olfr plays an integral role in kidney function [133,134]. It was speculated that IL-36R KO improved renal damage in UUO mice by regulating the enzymes important for collagen remodeling and blood pressure homeostasis. The molecular basis of IL-36R mediated regulation of Prss35, Olfr, and Vmnr needs to be further explored. Moreover, this work further implies that IL-36R may be involved in molecular pathways that influence fibrosis post kidney injury.

In summary, several studies have brought forth evidence that IL-36R signaling regulates fibrogenesis and may be a viable candidate for resolving fibrotic diseases in multiple organ systems. While the nature of the biological activities of IL-36/IL-36R signaling may be dependent on the tissue and cell type, it was noted that several similar pathways mediated by IL-36/IL-36R exist among various organ fibro-pathologies (Figure 3). Whether these shared pathways work individually or in concert to facilitate fibrosis progression is unknown. Thus, additional studies are needed to fully elucidate the immuno-mechanisms related to IL-36 stimulated fibrosis progression.

## 4. Conclusions and Future Directions

Organ malfunction due to fibrotic development has been associated with the end stages of many disorders such as pulmonary, renal, and myocardial disease [2,135,136,137]. The data reviewed here point to a possible link between the IL-36R pathway, inflammation, and fibrosis, and there is a growing amount of evidence that has implicated the IL-36R signaling pathway in fibrotic disease. The IL-36/IL-36R pathway in the context of fibrosis disease can be viewed as very complex and involves the interaction between many cells types and molecules that cooperate to allow for the progression of fibrosis. The information reviewed here sheds light on the possibility that the pathways involved may be shared among the various fibrotic diseases. Namely, the IL-36R regulation of (1) IL-17A associated fibrosis and (2) pro-fibrogenic actions of fibroblasts stood out as consistent features of fibrosis progression in multiple cases. Most interestingly, studies found that inhibiting the IL-36/IL-36R pathway prevented the fibrotic changes associated with numerous disease models, and thus provided strong evidence to support the significance of IL-36R in organ fibrosis. These data amplify the therapeutic potential of disrupting IL-36R signaling as a method to treat tissue inflammation and fibrosis. This new line of research has the potential to generate new therapeutic discoveries that can help prevent fibrotic disorders and the detrimental consequences that invoke organ dysfunction. Thus, understanding the mechanisms underlying IL-36R mediated tissue fibrosis will be essential to achieving this goal.

### Future Directions to Be Considered

The cell-specific roles of IL-36 cytokines and IL-36R in organ fibrosis: The data described in this review point to macrophages, DCs, fibroblasts, and T cells as some of the major cell types involved in fibrogensis. Therefore, employing cell type specific knockout mice would help define the individual cell specific roles of IL-36R and eliminate off target side effects that can occur upon globally knocking out IL-36R in all tissues and cells. Furthermore, these studies could identify the underlying mechanisms by which the pro-fibrotic actions of these cell types are modulated by the function of IL-36/IL-36R.

The role of IL-36R independent of immune-inflammatory response: Although the data have emphasized the role of an inflammation related mechanism, it would be beneficial to evaluate whether immune cell independent mechanisms are operational in IL-36R mediated fibrosis. For example, it would be interesting to understand how IL-36R influences other pathways in a setting in which the immune cell response was blocked or depleted. The discovery that IL-36R signaling is also functional in non-immune cells such as cardiomyocytes, renal TECs, fibroblasts, and affects cellular processes such as apoptosis, provides the rationale for exploring the immune independent roles of IL-36R more extensively. These studies would help better define the additional mechanisms that are responsible for eliciting the pro-fibrotic action of IL-36R in the setting of tissue injury.

The role of IL-36/L-36R in additional fibrosis models: As described in the above sections, the mechanism by which IL-36 mediates tissue fibrosis is complex and may differ depending on the tissue environment. To gain better insight into the mechanism, it would be beneficial to examine this pathway in additional experimental models of organ fibrosis such as liver fibrosis, brain glial scarring, myelofibrosis (bone marrow), and systemic sclerosis. Examining whether functions of the IL-36/IL-36R pathway impact the T cell response in these diseases states also needs to be addressed. In line with these studies, identifying the initiating factor that induces the IL-36/IL-36R signaling pathway would also be a relevant area of exploration.

Clinical relevance: Most of the studies related to the IL-36/IL-36R were performed in vitro or in animal models. To establish clinical relevance, it would be interesting to further explore how IL-36/IL-36R plays a role in human disease. For example, histological analysis of human fibrotic tissues and genetic screening for IL-36R polymorphisms may help identify biological modifications that impact an individual’s susceptibility to fibrosis under disease conditions. These studies could also validate IL-36R associated changes as general markers for fibrosis. Here, we highlighted work suggesting that the IL-36R antagonist, IL-38, may be considered an integral player in the pathways that drive fibrosis of tissues under pathological conditions. Additional studies to examine the profile and function of IL-38 in the context of human disease are also an interesting research area.

The therapeutic potential of IL-36R inhibition as a strategy to combat fibrosis: The current strategies to eliminate fibrosis include suppressing the deposition of ECM components, limiting the differentiation of resident cells to myofibroblasts, inhibiting IL-1 R family signaling, and blocking TGF-β induced fibroblast activation. However, these approaches are not commonly utilized as methods to eliminate fibrosis [2,4,5,6,7,8,9,10,11,12,13,18,54]. On the basis of recent data, inhibition studies have implicated the IL-36R receptor and its putative cytokines in tissue fibrogenesis in several organ systems; therefore, it would be favorable to carry out additional studies to validate the IL-36 KO as a feasible approach. One such experiment would include using an IL-36R inhibitor in a regression model of fibrosis to check its capacity for reversal of disease. These preclinical studies would provide interesting data that would establish the IL-36R pathway as a substantial player in fibrosis progression.

## Figures and Tables

**Figure 1 ijms-21-06458-f001:**
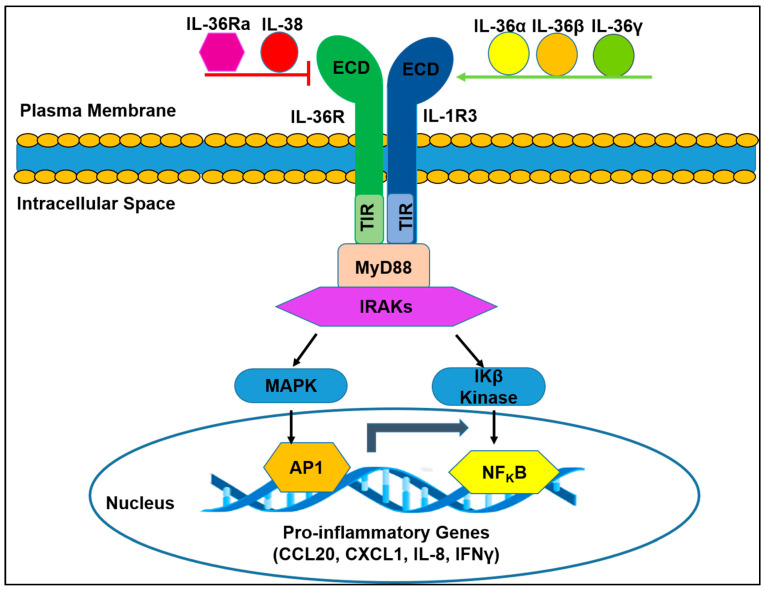
Structural and molecular depiction of interleukin 36 receptor (IL-36R) signaling cascade. Upon binding to its agonists (IL-36α, IL-36β, or IL-36γ), IL-36R heterdimerizes with its co-receptor IL-1R3 (IL-1RAcP). The formation of the IL-36R/IL-36/IL-1R3 complex results in the recruitment of MyD88 and IRAKs, which subsequently activates the MAPK and NFKB pathways. Activation of MAPK results in the transcription of inflammatory genes regulated by AP-1, while the phosphorylation of IKβα (a negative regulator of NF_K_B) by Ikβ kinase leads to the translocation of NF_K_B to the nucleus, which aids in the transcription of genes important for mediating inflammation. The IL-36R mediated activation of pro-inflammatory gene expression can be inhibited by IL-36Ra or IL-38. Extracellular domain (ECD), toll/interleukin-1 receptor domain (TIR), myeloid differentiation primary response 88 (MyD88), interleukin-1 receptor associated kinases like 1 and 2 (IRAKs), mitogen-activated protein kinase (MAPK), activator protein-1 (AP-1), NF-Kappa-B Inhibitor Alpha (IKβα), and nuclear factor kappa light chain enhancer of activated B cells (NF_K_B), CCl, C-C motif chemokine ligand; CXCL, C-X-C motif chemokine ligand; IFN, interferon.

**Figure 2 ijms-21-06458-f002:**
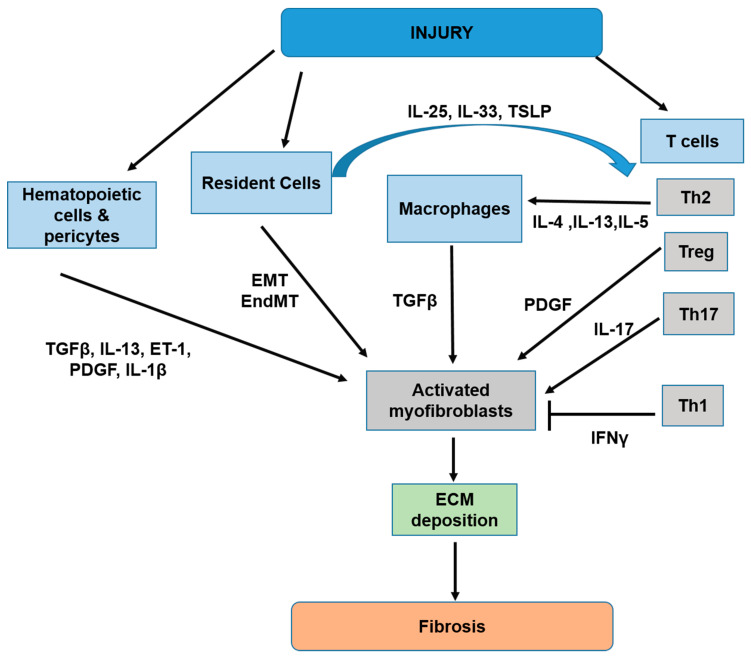
Major contributions of immune and non-immune cells in tissue fibrosis. During fibrosis, macrophages and T cells promote fibrosis via activation of myofibroblasts and release of pro-fibrotic inflammatory factors. Non-immune cells can also be transformed into “myofibroblast-like” cells to produce collagen factors that advance the progression of fibrosis. TGF, transforming growth factor; PDGF, platelet-derived growth factor; EMT, epithelial to mesenchymal transformation: EndMT, endothelial to mesenchymal transformation; ECM, extracellular matrix; TSLP, thymic stromal lymphopoietin; Endothelin-1 (ET).

**Figure 3 ijms-21-06458-f003:**
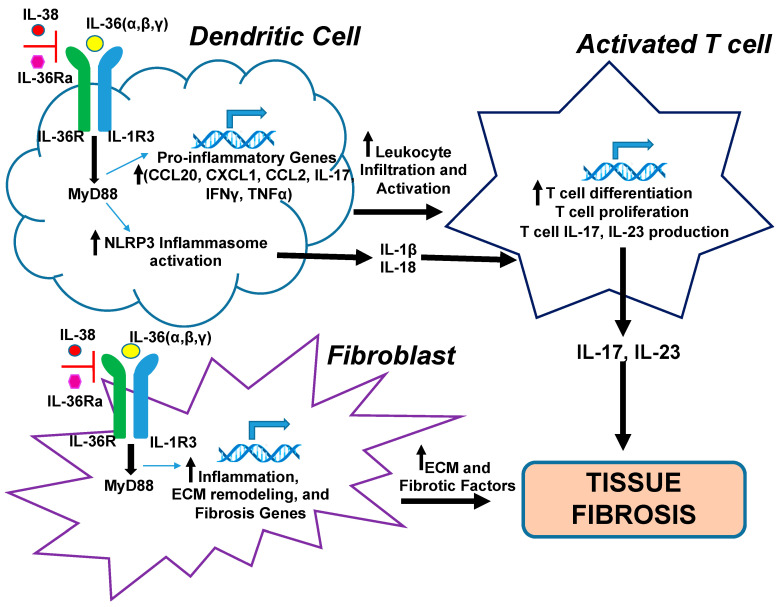
Potential molecular mechanisms of IL-36R induced tissue fibrosis. IL-36R signaling promotes fibrosis development in tissues by (1) regulating dendritic cell and T cell immune responses through a MyD88 dependent pathway and (2) driving inflammation, extracellular matrix (ECM), and fibrogenic gene upregulation in fibroblasts. Nod-like receptor family pyrin domain containing 3 (NLRP3).

**Table 1 ijms-21-06458-t001:** Summary of interleukin (IL)-36/IL-36 receptor (IL-36R) mediated tissue fibrosis.

Organ	Disease Model	IL-36R Ligand Involved	IL-36 Producing Cells	Possible Downstream Targets	Outcome of IL-36R Inhibition	Refs
Lung	Bleomycin induced IPF	IL-36γ	IL-17A expressing Macrophages (CD9^hi^, IL3γ^+^)	Th17 T cells	IL-38 overexpression reduced lung inflammation, IL-17A production, and fibrosis	[88,90]
Kidney	UUO	IL-36α	Renal TECs	BMDCs, Th17 T cells	IL-36R KO prevented T cell activation/infiltration, and suppressed the occurrence of renal TILs	[21]
Heart	LAD coronary artery ligation model of MI	IL-38	Cardiomyocytes	DCs	Recombinant IL-38 treatment protected against cardiac fibrosis and cardiomyocyte apoptosis. IL-38 also altered DC differentiation and T cell responses	[89]
Intestine	DSS and TNBS induced intestinal colitis and fibrosis	IL-36α	CD14+CD64+ CD163^+^ Macrophages	α-SMA^+^ Fibroblasts	IL-36R KO protected mice from colitis and fibrosis. IL-36R deficiency also reduced collagen production and blunted the presence of activated fibroblasts (CD90^+^, α-SMA^+^)	[87]
Pancreas	Chronic Pancreatitis	IL-36α	--------	Myofibroblasts	---------	[120]

Bone marrow-derived dendritic cells (BMDCs); dendritic cells (DCs); dextran sulfate sodium salt (DSS); idiopathic pulmonary fibrosis (IPF); knockout (KO); left anterior descending (LAD); myocardial infarction (MI); T helper 17 (TH17); 2,4,6-trinitro benzene sulfonic acid (TNBS); tubular epithelial cells (TECs); tubulointerstitial lesions (TILs); unilateral ureteral obstruction (UUO).

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
