# Peer review of "Interleukin-36 Cytokine/Receptor Signaling: A New Target for Tissue Fibrosis"

_ijms, 2020, doi:10.3390/ijms21186458_

Round 1

Reviewer 1 Report

This is a comprehensive review on the involvement of IL-36 in fibrosis.

  1. It would be easier to understand this involvement if chapter 4.1 "Overview of tissue fibrosis development" would appear first and only then the various  pathological situations in the various organs would be described (repetition excluded) and followed by Table 1.
  2. On top of Fig.2 a scheme of fibrosis development would help a lot including insertion of the various cytokines involved (IL-1, IL-18, IL-33 IL-38, TGF-b, TNF etc.).
  3. Where is IL-36 thought to be located in the hierarchy of the IL-1 family. Please elaborate?
  4. Row 284: Reference clinical trials?
  5. There is a very recent review on IL-36 in chronic inflammation by Neurath MF that should be referenced.
  6. Row 161: "of" should be ommitted.
  1.  

Author Response

Response to the comments of Reviewer1

Comment 1:It would be easier to understand this involvement if chapter 4.1 "Overview of tissuefibrosis development" would appear first and only then the variouspathological situations in the various organs would be described (repetition excluded) and followed by Table 1.

Response: We have reorganized the relative contents according to the reviewer’s suggestion. Section 4hasbeen moved to Section 3.1(pages 5-7, lines 341-486). The original section 3 isnow section 3.2followed by Table 1. The originalsection 4.2 is now section 3.3.We also eliminated repetitive text in section 3.3.

Comment 2:“On top of Fig.2 a scheme of fibrosis development would help a lot including insertion of the various cytokines involved (IL-1, IL-18, IL-33 IL-38, TGF-b, TNF etc.)”

Response: To address the reviewer’s suggestion, we generated a new scheme (Figure 2) to summarize the major mechanismsinvolved in the fibrosis development, including the major types of the cells and the associated cytokines that contribute to fibrogenesis. We also added details explaining these mechanisms in the text (section 3.1, page 6, lines 352-383).The corresponding references have also been added.We also modified the figure 2 (new Figure 3) to reflect the involvement of IL-1β, IL-18, and TNFα in mediating T-cell activation.

Comment 3:“Where is IL-36 thought to be located in the hierarchy of the IL-1 family. Please elaborate?”

Response: we added additional background content and references to provide details about the IL-1 cytokine and IL-1 Receptor families and their hierarchy(page 2,lines51-61).

Comment 4: “Row 284: Reference clinical trials?”

Response: To address this comment, we added Neufert’s 2020review as referencefor the Ulcerative colitisclinical trials (reference #119, Line 645). Given that the results from this study have not been formally reported yet, we also cited the website where the clinical trial status can be monitored https://clinicaltrials.gov/ct2/show/NCT03482635(page 9, lines 645-646

Comment 5: “There is a very recent review on IL-36 in chronic inflammation by Neurath MF that should be referenced”

Response: we added the suggested Neufert etal2020 as reference to section3.2(reference # 119, Line 645).

Comment 6: “Row 161: "of" should be omitted.”

Response: This has been corrected.

Reviewer 2 Report

Tissue fibrosis is a major unresolved medical problem, which impairs the function of  various tissues. The molecular mechanisms involved in this process are poorly understood which hinders the development of effective therapeutic strategies. Therefore, the review “Interleukin-36 cytokine/receptor signaling: a new  target for tissue fibrosis” presented by Elaina Melton and Hongyu Qiu for publication in the International Journal of Molecular Sciences is very relevant.

The emerging evidence from data presented in this review  indicate  that interleukin 36 (IL-36) and the corresponding receptor (IL-36R), a newly-characterized  cytokine/receptor signaling complex involved in immune-inflammation, plays an important role in  the pathogenesis of fibrosis in multiple tissues. In this review is highlighted the most recent works that examine the fibrogenic role of IL-36/IL57 36R in several models of disease such as lung, kidney, cardiac, intestinal, and pancreatic fibrosis. Authors also summarize the new insights in the potential of IL-36R inhibition as a therapeutic strategy to combat pro-fibrotic pathologies.
The review presents quite informatively molecular basis and biological function of IL-36R in cells. This section of the review includes submission about IL-36R molecular structure, IL-36R function-associated molecules, and distribution and biological functions of IL-36R.

Of particular interesting is the section dealing with the pathological role of IL-36R. Here, the participation of IL-36R in fibrosis is demonstrated using the example of various tissues and organs. It should be noted that this section cites very impressive studies indicating the protective effect of IL-36R inhibitors on the development of fibrosis in various organs. 
The review discusses in some detail cellular and molecular mechanisms of tissue fibrosis and IL-36 / IL-36R signaling (section 4). But I would recommend the authors to change this section 4 with section 3, then they can remove some of the repetitions that are found in these sections.In general, the review is very informative, colorfully illustrated and will be useful both for narrow specialists studying the mechanisms of fibrosis and for a wide range of readers.The particular value of the review should be emphasized in that the authors indicate the perspectives for new research in the study of this new mechanism of fibrosis. Although the mechanism of interleukin-36 cytokine/receptor signaling  is far from being fully understood, the authors are optimistic that inhibitors of IL-36R may be very effective for prevention and treatment of  tissue fibrosis.              

Such a forward-looking review should be published.

Author Response

Response to the comments of Reviewer2

Comment 1:The review discusses in some detail cellular and molecular mechanisms of tissue fibrosis and IL-36 / IL-36R signaling (section 4). But I would recommend the authors to change this section 4 with section 3, then they can remove some of the repetitions that are found in these sections

Response: We thank the reviewer for this comment. We have reorganized the relative contents of Sections 3 and 4according to the reviewer’s suggestion. Section 4hasbeen moved to Section 3.1(pages 5-7, lines 341-486). The original section 3 isnow section 3.2followed by Table 1. The originalsection 4.2 is now section 3.3.We also eliminated repetitive text in section 3.3.In addition, we have added a new figure (Figure 2 ) to summarize the contents of section 3.1 ( the original section 4)